# Influence of Micro-Pore Connectivity and Micro-Fractures on Calcium Leaching of Cement Pastes—A Coupled Simulation Approach

**DOI:** 10.3390/ma13122697

**Published:** 2020-06-13

**Authors:** Janez Perko, Neven Ukrainczyk, Branko Šavija, Quoc Tri Phung, Eddie A. B. Koenders

**Affiliations:** 1Institute for Environment, Health and Safety, Belgian Nuclear Research Centre SCK CEN, Boeretang 200, 2400 Mol, Belgium; quoc.tri.phung@sckcen.be; 2Institute of Construction and Building Materials, Technical University of Darmstadt, Franziska-Braun-Str 7, 64287 Darmstadt, Germany; ukrainczyk@wib.tu-darmstadt.de (N.U.); koenders@wib.tu-darmstadt.de (E.A.B.K.); 3Faculty of Civil Engineering and Geosciences, Delft University of Technology, Stevinweg 1, 2628 CN Delft, The Netherlands; b.savija@tudelft.nl

**Keywords:** numerical models, pore scale, cement leaching, micro-fractures, pore connectivity, effective diffusivity

## Abstract

A coupled numerical approach is used to evaluate the influence of pore connectivity and microcracks on leaching kinetics in fully saturated cement paste. The unique advantage of the numerical model is the ability to construct and evaluate a material with controlled properties, which is very difficult under experimental conditions. Our analysis is based on two virtual microstructures, which are different in terms of pore connectivity but the same in terms of porosity and the amount of solid phases. Numerical fracturing was performed on these microstructures. The non-fractured and fractured microstructures were both subjected to chemical leaching. Results show that despite very different material physical properties, for example, pore connectivity and effective diffusivity, the leaching kinetics remain the same as long as the amount of soluble phases, i.e., buffering capacity, is the same. The leaching kinetics also remains the same in the presence of microcracks.

## 1. Introduction

The durability of concrete-made civil engineering and cultural heritage structures predominantly depends on environmental factors like pollution and climate change [1], which initiate a wide variety of decay processes [2,3]. Many of these processes, e.g., calcium leaching, carbonation, alkali silica reaction, and delayed ettringite formation, are based on leaching and/or precipitation. As the basis of these processes is the leaching of primary minerals and formation of secondary minerals, it is important to understand each of these processes separately.

Leaching of cementitious materials is a dissolution–diffusion process of ions presented in the pore solution. Among this, Ca-leaching commonly occurs as most of the phases in cementitious materials contain Ca. Leaching of calcium is considered one of the most detrimental degradation processes in reinforced and non-reinforced concrete structures, of which the service life is expected to be very long, such as hydraulic structures (e.g., dams, water towers) or nuclear waste disposal facilities (e.g., surface disposal, tunnel lining concrete for deep disposal) [4]. Under normal leaching conditions, which are typically without advection, the calcium leaching of concrete is extremely slow due to the diffusion-controlled process. However, the leaching might significantly alter a variety of cement matrix properties including the increase in porosity (both capillary and gel porosity) [5], decrease in pH [6], increase in transport properties (e.g., diffusivity and permeability) [7,8,9], and detrimental effects on mechanical properties related to changes in phase fraction [10,11]. Considering the extremely slow leaching process but the need to study the performance of concrete in the long-term, the use of accelerated testing methods is considered a relevant approach for better understanding the leaching phenomena and its effects on the alteration of microstructure, mineralogy, and transport properties. With the help of accelerated experiments, it is also possible to verify prediction models. A variety of accelerated experimental methods have been proposed to study the leaching including the application of an electrical field [12] or flow-through conditions [13], the use of deionized water [14,15] or low-pH solutions (e.g., acidic environment) [16,17], or using high-concentration solutions [18,19] (which not only lowers the pH but also increases the solubility of Ca). Among these, using an NH_4_NO_3_ solution is one of the most popular methods to accelerate the Ca-leaching process. This method not only results in faster leaching kinetics under diffusive-transport conditions compared to other mentioned methods but also has the same end-products. Instead of performing accelerated experiments, in this study, we performed an accelerated numerical leaching following the approach proposed in a previous study [20] to reduce the computational time.

It is commonly assumed that the leaching would change cementitious materials by coarsening the microstructure and increasing the transport properties as discussed above. This assumption is supported by various studies in the literature. However, there are still not many quantitative studies that allow us to answer the question of to what extent the microstructure, mineralogy, and transport properties would be altered. For example, most studies show an increase in porosity and coarser pore size distribution for leached materials using ammonium nitrate solution. However, the extent of alterations strongly depends on the compositions of the samples, especially the types of cement and supplementary cementitious materials added. As a general trend, the samples made from the cements that produce less portlandite content during hydration exhibit a better leaching resistance [10]. The same trend is observed for materials having pozzolanic activity [21,22] because the pozzolanic reactions reduce the portlandite content. Berra et al. [21] showed that an addition of 3.8% nanosilica could slightly prevent the increase in porosity due to the accelerated leaching of cement paste CEM I. Despite such a slight effect on porosity, the decrease in compressive strength of the leached sample is significantly reduced. In some cases, e.g., the study of Segura et al. [22], the porosity even decreases for CEM III (blast-furnace slag) and CEM IV (fly ash +limestone filler) mortar samples due to precipitation in the first few days of leaching. Nevertheless, if the leaching process is long enough (more than 2 weeks), porosity still increases. The maximum porosity increase does not depend on the water/cement ratio and could increase up to about 30% for CEM I mortars after 32 days of leaching. It is interesting to observe that the pore surface area significantly increases, without significant modification in threshold pore diameter and critical pore diameter for the cases of CEM III and CEM IV mortars. This perhaps occurs at the gel level that is not captured by the measurement techniques to identify the threshold and critical pore diameters. Nevertheless, the critical pore diameter and threshold pore diameter are increased for the cases of CEM II and CEM I mortars, respectively. Cheng et al. [23] also found that the addition of slag to cement mortar would help to reduce the alteration in porosity due to leaching. As leaching is a diffusion-controlled process, the porosity modification is expected to be a function of degraded depth, as shown in the study of Poyet et al. [24]. At the surface directly in contact with the NH_4_NO_3_ solution, the porosity is significantly increased from 26% to 80% for cement paste with a water/cement ratio of 0.5 and 18.5% fly ash addition [24]. The addition of silica fume helps to improve the resistance of concrete to leaching, as shown in the study of Bibi et al. [8]. The concrete samples with a silica fume content of 8% have shown significant improvement in strength and more resistance to the NH_4_NO_3_ attack compared to the samples without silica fume addition. The increases in porosity and coarser microstructure would result in an increase in transport properties. With the tricalcium silicate pastes leached in ammonium nitrate solution, the chloride diffusivity of leached pastes could increase by a factor of 1.8 to 3.6 depending on the alteration level [25]. On the other hand, the dissolved gas diffusivity of leached cement (CEM I) paste samples might increase more than one order of magnitude depending on the defusing gases and initial microstructure of the materials, as shown in the study of Phung et al. [26].

Compared to leaching in ammonium nitrate solution, using deionized water in the leaching study is limited due to the long experimental time and insufficient degraded materials obtained within the reasonably experimental duration to perform necessary characterizations. Jain et al. [15] investigated the leaching in deionized water of three types of cement pastes with the same water to cement ratio (w/c) of 0.4 including plain CEM I/II pastes, cement paste with 6% silica fume replacement, and pastes with 10% fly ash and glass powder replacements. The same trend for the porosity increases as leaching in NH_4_NO_3_ is observed for all the cement pastes, and the longer the leaching duration, the higher the porosity increase. The plain pastes exhibit the highest porosity increase, up to 11% after 90 days of leaching in deionized water, which is still much lower compared to leaching in ammonium nitrate solution. The modified pastes with glass powder show the lowest porosity increase of only 5%. This is attributed to the presence of NaOH in its pore solution resulting in a reduction in portlandite solubility. The modified pastes with silica fume/fly ash also exhibit a similar porosity increase of about 8%. Porosity increase is mainly caused by the dissolution of portlandite, which creates the capillary pores. However, C-S-H leaching also contributes to increase the porosity for a long leaching duration, which increases both gel and capillary pore volume. After 90 days of leaching in deionized water, Jain et al. [15] showed that C-S-H could contribute almost 3%, while portlandite contributed 8% to the total porosity increase. The contribution of C-S-H is clearer when leaching in ammonium nitrate solution, as reported by Phung [27]. In the case of leaching performed in ion-exchanged water and using a high w/c ratio of 0.7, the Ordinary Portland Cement (OPC) cement paste could increase the porosity up to 22% after 56 weeks of leaching, as shown in the study of Haga et al. [28]. In that case, by comparing the measured total porosity and the calculated porosity increase resulting from portlandite leaching, the leaching of C-S-H is also observed and contributes to about a 6% porosity increase. The pore size distribution also shows a pore volume increase in the pore size ranging from 0.05 to 0.5 µm after leaching for 56 weeks. 

In summary, it can be seen from the above discussion that there are many factors affecting the Ca-leaching of cement-based materials. Besides the leaching conditions (e.g., in ammonium nitrate solution, deionized water), the composition of materials (e.g., cement types, addition of supplementary cementitious materials), which defines the microstructure and, therefore, transport properties (diffusivity and permeability), is utmost important. Transport properties determine how fast Ca ions are transported from the cement matrix to the surrounding environment. The diffusion coefficient is basically influenced by the accessible porosity, connectivity [9], and the presence of micro-fractures in the matrix, which will be intensively discussed in this study.

Modeling of the leaching process has also been pursued by many researchers to interpret the experimental results and predict the properties of leached materials, which is hard to obtain by experimental study. Most of the models focus on the prediction of the leached depth (similar to carbonation depth) of leached materials and characterize its mechanical behavior [29,30,31,32]. However, there are only a few studies that discuss the modification of the microstructure after leaching [24,33,34] and the effects of leaching on transport properties [2,7,35,36]. Nguyen et al. modeled the effects of Ca-leaching on the mechanical properties of concrete [32], while Bentz et al. considered the influence on the microstructure and diffusivity [37]. A 2D model was developed by Mainguy et al. to predict the degraded depth and the variation in leached calcium flux with time [31]. Jacques et al. developed a thermodynamic model to calculate the geochemical changes of concrete during leaching with rain and soil water types [38], while Yokozeki et al. modeled the long-term behavior of cementitious materials used in an underground environment [39]. Comparison of different modeling approaches for Ca leaching has been performed in [40]. These models are mainly based on the assumption of thermodynamic equilibrium of calcium in the solid and liquid phases in pure/deionized water, which was first studied by Berner [41,42], of the solid–liquid equilibrium curve of calcium in ammonium nitrate solution [43] for the cases of accelerated leaching in NH_4_NO_3_ solution. Recently, Wan et al. developed a solid–liquid equilibrium curve of calcium in ammonium nitrate solution [43], which was later used in his model [44].

The effective diffusivity is a crucial parameter for concrete durability, which is typically simulated from virtual microstructures generated by conventional growing sphere cement hydration models, generally overestimating experimental data by up to a factor of ten [45,46]. The main reason for this is that the connectivity (or percolation) of the pore network generated by growing sphere models exhibits a sharp de-percolation threshold, i.e., a particular porosity whereby the volume fraction of connected capillary pores in a cement paste decreases to zero, which happens at about 4% porosity [45,46,47]. In other words, the pores of sizes larger than 1 μm become connected at porosities above 4%. In a real material, a gradual de-percolation effect already starts at 40% porosity, and reaches the point where half of the pore volume is being disconnected at the total porosity of about 24% [48,49]. The capillary porosities at which de-percolation starts and ends are two important critical points, as between these points, the dominant transport pathway changes from a connected capillary network, i.e., parallel capillary-C-S-H pores, into a serial connection of capillary pores, built-up from C-S-H gel pores. In this paper though, an improved growing sphere modeling approach has been proposed for the morphological description of the hydrated cement paste. With this, an overestimation of effective transport properties has been avoided while maintaining the effectiveness of the conventional particle algorithms.

Nowadays, direct modeling of the morphological nature on effective transport is widely used for calculating the average transport properties at different scale levels [45,47]. However, direct coupling between the pore chemistry and transport through a porous material still remains largely underdeveloped but shows encouraging potential. Modeling of mineral dissolution at the pore scale has been studied in recent years by different numerical approaches. A benchmark between different modeling approaches shows good agreement between results [50]. In particular, the lattice-Boltzmann method has been extensively used for the pore-scale modeling of mineral dissolution [51,52,53] because of its computational efficiency and ability to deal with high-concentration gradients and contrasting material properties [54] associated with dissolving geochemical systems. 

In this paper, an advanced reactive-transport model is provided where results are used for leaching-induced aging, applied to virtual cementitious microstructures generated by the Hymostruc model [55]. In these simulations, emphasis is on the effect of pore connectivity and its subsequent impact on leaching. Firstly, the pore connectivity of the original Hymostruc model (i.e., reference microstructure) was modified by implementing an innovative hollow shell growing sphere approach (i.e., modified microstructure), which reduced the connectivity to a more realistic one. Secondly, both microstructures, i.e., a reference one and a modified one, were numerically cracked by introducing mechanical fracturing simulations. This was done via a coupling with the Delft lattice model. Lastly, chemical leaching was calculated on four resulting microstructures (two references and two cracked ones) by using a pore-scale reactive-transport model, and the results were compared to see the effect of pore connectivity on leaching, and with this, on the ageing performance of cementitious materials.

## 2. Physical Framework 

### 2.1. Microstructure of Hydrated Cement Paste

A hydrated cement paste considered in this paper mimics an Ordinary Portland Cement (OPC) and water mixture with a mass ratio of w/c = 0.5, cured for a hydration period of 28 days, while achieving 15.7 vol.% of capillary porosity, 3.44 vol.% clinker, 50 vol.% calcium silicate hydrates (C-S-H), 10.6 vol.% portlandite, and 20 vol.% aluminates [56]. This particular cement paste composition was used to represent the experimental results from [56], which turned out to be a valuable reference for the measured effective diffusion coefficient. The chemical degradation process described in this research is based on calcium leaching from an Ordinary Portland Cement (OPC, i.e., CEM I). The proposed modeling approach can be used to simulate the effects of other microstructural parameters, e.g., w/c, chemical composition, mineral additions (i.e., different cement types), chemical additives, etc.; however, we believe this would not change the conclusions of this paper. The porous microstructure of cementitious materials spans over multiple length scales, ranging from nanometer-sized gel pores inside hydration products, via micrometer-sized capillary pores in-between the reactive cement grains, toward millimeter-sized air bubbles. As the pore structure develops with hydration time, the different types of pores are associated with the local microstructural domains. The capillary pores embody remnants of the initial water phase that was not overgrown by the so-called outer hydration products precipitating in the interstitial space between the cement grains. The hydration products precipitated inwardly; inside, the initial boundaries of the cement particles are called inner products. However, a considerable number of the cement particles also exhibit inner pores in the same size range as the capillary pores, and these porous particles are called hollow-shell hydration particles [57]. Quantitative investigations of hollow-shell cement particles are scarce due to a lack of suitable methods for the microstructural characterization of the hollow shell porosity, but electron nano-tomography clearly demonstrated its significance. Holzer et al. [57] showed experimentally by a focused ion beam scanning electron microscope (FIB-SEM) that the capillary porosity surrounded (closed) by C-S-H gel pores (here, termed as hollow shell or closed porosity) comes from the hollow shell mode of hydration. However, thus far, the effects of hollow shell porosity formation have not yet been investigated in terms of transport and durability properties.

### 2.2. Mechanical Degradation

As the cementitious microstructure is subjected to external (mechanical) or internal (shrinkage) loading, cracks can arise in the microstructure due to its low tensile strength and strain capacity. Apart from the loading condition, a possible connectivity of the cracks largely depends on the crack morphology and size of the cracks in a loaded microstructure. More specifically, the capillary porosity and associated pore structure, as well as the amount of hydration products and their spatial distribution, govern the fracture process at the microscale level. The evolution of microcracks has been reported to have a significant effect on the transport properties of cementitious materials [58,59]. This is also why the effect of microcracking on the leaching behavior of two simulated microstructures was considered explicitly. With the Delft lattice model (TU Delft, The Netherlands), cracks were introduced in a selection of virtual microstructures by imposed uniaxial tensile loading. This was done as, in general, mode I is considered to be the dominant mode of cracking in cementitious materials [60].

### 2.3. Chemical Degradation

The chemical degradation process described in this research is based on a microstructure representing an Ordinary Portland Cement (OPC). Calcium leaching is a process in which Ca-containing solid phases (e.g., portlandite, C-S-H) of cement-based materials dissolve because the solid phases are not in equilibrium with its environment. Due to the evolving chemical gradients, ions are transported through the pore fluid to the external environment. Leaching of calcium ions (along with other ions, such as silica ions) is a very important degradation mechanism as calcium is the major constituent of various hydrated cement paste mineral phases such as portlandite, calcium silicate hydrates, and aluminate phases, which are the main building blocks of a cement paste matrix. Calcium leaching by nature is a very slow process where the leaching front extends only a few millimeters every hundred years [61]. Due to this, modeling or accelerated experiments are frequently used to understand the effects of leaching on concrete structures and its consequence for the long-term durability. 

## 3. Modeling Framework

### 3.1. Microstructure Generation

The hydrated virtual cement paste microstructures were simulated by the Hymostruc model representing the experimental results from [56] and are summarized in Table 1. The model Hymostruc is the acronym for HYdration, Morphology, and STRUCtural development. The initial state of the microstructures is determined by a random positioning of the un-hydrated cement particles, using a non-overlapping algorithm. A water-to-cement mass ratio of 0.5 is used, whereas the particle size distribution follows the Rosin–Rammler distribution Equations (1) and (2), representing a specific surface Blaine value of 400 m^2^/kg, and thus including a range of particle diameters from 1 to 38 μm, as a good numerical compromise according to [47]. The Rosin–Rammler distribution is defined as:(1)$$G\left(d\right)=1-e^{-b\;d^{n}}$$
where *G*(*d*) is the cumulative weight (in g) of the particles with diameter (in μm), and parameters *n* and *b* are constants defined by the fineness of the cement. The number of particles in a certain fraction is calculated from the weight per fraction, obtained by differentiating Equation (1) with respect to the particle diameter *d*, as well as by dividing this by the specific density of the cement (*ρ*) and by the volume of a single particle (*V*_p_ = πd^3^/6):(2)$$N_{d}=\frac{b\;n\;d^{n-1}e^{-b\;d^{n}}}{\rho \;V_{p}}$$

After generation of this initial particle structure, hydration algorithms were employed to simulate the stepwise evolution of the particle hydration process and associated expansion of the outer shells of hydration products, as well as the growth of portlandite particles. The number of portlandite particles are placed in the simulated pore volume, which is calculated according to a stepwise approach that corresponds to the volumes of the silicate reactions [62]. The maximum total number of portlandite particles in the simulated volume of 100^3^ μm^3^ is fixed to eight thousand. The outer shell volume of C-S-H (including aluminate hydrates C-A-H) produced by a single particle is precipitated around the same particle, while the crystalized portlandite is distributed in random proportions forming the portlandite (calcium hydroxide (CH)) particles. The volume changes were obtained by considering a volume balance approach, which is based on the clinker phases’ hydration stoichiometry. For cement reactions, the following reaction equations are considered:(3)$$C_{z}S+\left(z-\frac{C}{S}+y\right)H\;\to C_{\frac{C}{S}}{\mathrm{SH}}_{y}+\left(z-\frac{C}{S}\right)\mathrm{CH}$$
where z = 2 or 3 and represents the silicate reaction for C_3_S (alite or tricalcium silicate) and C_2_S (belite or dicalcium silicate), respectively. As the leaching modeling approach in this paper is based on the thermodynamics of Ca and Si speciation only, the contribution of aluminate hydrates (C-A-H) was indirectly considered, by increasing the relative amount of formed C-S-H. This was implemented by slightly increasing the apparent Ca/Si molar ratio of the C-S-H (above the nominal 1.8 value), to increase the C-S-H/CH volume ratio.

In the hydration model Hymostruc, the formation of a 3D microstructure is simulated with the assumption that hydrating cement particles behave like expanding spheres, which follow an integrated kinetics approach according to the so-called ‘Basic Rate Equation’ concept [55], which describes the reaction front as an individual reacting particle *i* with diameter *d*, at time *t*:(4)$$\frac{\Delta \delta_{d,i}}{\Delta t}=K_{0}ΩF_{Arr}{\left(\frac{\delta_{tr}}{\delta_{d,i}}\right)}^{\beta \;\lambda }$$

In this kinetic Equation (4), *K_0_* is the initial rate of hydration reaction of the individual cement particles, *F*_Arr_ is the Arrhenius function for temperature dependence, and the Ω factor accounts for the state of water in the microstructure during hydration [47,55]. Parameters *δ_tr_* and *δ_d,j_* are the transition thickness (a constant threshold value) and the thickness of the total layer of produced hydration products, respectively. The last part in Equation (4) uses a Boolean parameter λ, which accounts for the change in the particle reaction mechanism from a phase boundary (λ = 0) to a diffusion-controlled boundary (λ = 1), and *β* is an empirical constant [47,55]. The incremental evolution of the degree of hydration of cement is then calculated for each individual particle (and for each of the Bogue phases), and is based on the actual depth of the reaction front of a particle *i* (*δ_d_*), while Δ*δ_d_* is the incremental increase in this front. The Δ*δ_d_* is calculated for each reaction scheme, and the actual reaction front of a particle is obtained by mass averaging the individual reaction contributions. For a more detailed description of the algorithm, a reference is made to [63].

Particle-based cement hydration models typically overestimate the effective transport properties of microstructures, because the connectivity of the porous network is too high [45,48]. The original growing sphere (Hymostruc) model has been adjusted to make a better representation of a real microstructure of hardened cement paste. In this respect, a hollow shell mode of hydration (inspired by [57]) was implemented, where the inner hydration products were replaced by a closed (hollow shell) volume of capillary porosity, surrounded by small C-S-H gel pores, with the solid volume of the inner C-S-H being re-distributed into the outer shell C-S-H. This adjustment enabled a reduction in the actual connected capillary pore volume in the system, and the results of this calibration are detailed in the section describing microstructure generation. Two virtual microstructures are generated, which are different in terms of pore connectivity but the same in terms of porosity and the amount of solid phases. The reference microstructure is named H0, indicating that the hollow shell algorithm is not invoked by setting the value of the threshold particle diameter to 0 µm, as hollow shells are being created for cement particle sizes below this critical value. The second microstructure is modified for particles smaller than the critical diameter, calibrated to 20 µm here [57] and thus named H20, to provide a good match between the calculated pore de-percolation value with real material. In the newly proposed hollow shell algorithm, the volume of the inner C-S-H hydration products inside the particles was relocated to the outer shell (inspired by 3D nanotomography measurements [57]), and the resulting vacant space was considered capillary pore space, representing the hollow shell approach. This was implemented in the existing Hymostruc kernel following two major steps. First, the phase tag associated with the inner C-S-H hydration products was changed to a capillary pore (i.e., color changed from red to blue). Secondly, the implemented algorithm that accounts for the calculation of the expanded C-S-H growth around the particles was extended with the additional volume coming from the inner C-S-H shell volume, which is equal to the volume of the reacted cement. It was not necessary to change the overlap algorithm of the Hymostruc model, as it implicitly calculates the expansion of particle shells, also when considering hollow shell transformations.

### 3.2. Mechanical Degradation

Mechanical degradation of the cement paste microstructure is simulated using the Delft lattice model [64,65]. The Lattice model is a type of discrete element model in which the continuum is discretized by a set of simple beam or truss elements. In the past, these models have shown excellent capabilities for simulating deformation and fracture processes in quasi-brittle materials like cement paste and/or concrete [66], but also [67,68] nuclear graphite [69,70], and rocks [71,72] could be simulated very well. In the Delft lattice model, the solid material is discretized as a set of Timoshenko beam elements [73], which can transfer normal forces, shear forces, and bending and torsional moments [74]. Typically, each element is assigned with a linear ideal elastic brittle behavior. Loading is stepwise applied to the global mesh, and once an element becomes critical, meaning the element has reached the highest stress-to-strength ratio, it is identified and removed from the mesh. This procedure is then repeated until a certain force or displacement criterion has been reached. This is commonly referred to as a sequentially linear solution procedure [75]. As this approach closely mimics the process that obviously occurs during fracturing of quasi-brittle materials, very realistic global and softening behavior of crack propagation paths can be simulated. The network of beams inside a digitized microstructure is created by, first, the random placing of nodes within a sub-cell of each voxel and connecting them with beam elements, using the Delaunay triangulation [76] (Figure 1a). The heterogeneous nature of hardened cement paste systems is considered by employing a particle overlay procedure (Figure 1b). 

Simulating the fracture process in hardened cement paste systems was done by assigning elastic modulus and strength properties to each individual element. The elastic modulus of lattice element *i*-*j*, connecting the nodes *i* and *j*, is assigned by using a series (Reuss) model [78]:(5)$$\frac{2}{E_{i-j}}=\frac{1}{E_{i}}+\frac{1}{E_{j}}$$
where $E_{i}$, $E_{j}$, and $E_{i-j}$ are Young’s moduli for the phases i and j, and the stiffness of a lattice element connecting nodes i and j. Strength properties of each lattice element are assigned by the lowest one of the two connected phases, expressed mathematically as:(6)$$f_{t,i-j}=\min\left(f_{t,\;i},f_{t,j}\right)$$
where $f_{t,\;i}$, $f_{t,\;j}$_j_, and $f_{t,\;i-j}$ are the tensile strengths for phases i and j, and for a lattice element connecting nodes i and j, respectively. The elastic moduli of individual hydration phases in hardened cement paste have been measured by employing nanoindentation experiments [79,80]. In addition, the strength properties, although theoretically related to the indentation hardness, has not been measured directly for each individual phase. Therefore, results from [81] were applied, which were based on an inverse analysis to determine the strength of individual hydration phases. The values used as input for the current work are listed in Table 2.

The two generated lattice-based microstructures, i.e., H0 and H20 (the number 0 or 20 represents the size of the critical particle below which the hollow shell mode of expansion algorithm was applied, as described in the previous section), have both been subjected to a simulated uniaxial tension test. For this, one side of the microstructure was completely fixed, while the opposite side was subjected to a prescribed displacement. This configuration is schematically shown in Figure 2.

As the leaching analysis requires a voxel-based input, the deformed and fractured lattice configuration must be converted back to a voxelized microstructure after the stepwise Lattice analysis has ceased. This backward voxel conversion is done according to the following procedure; after first crack occurrence somewhere in the microstructure, the cracked elements in the lattice are identified; from there, it is checked whether the width of a crack exceeds the predefined threshold crack width; if that is the case, the voxels corresponding to the cracked lattice beams are converted into a pore voxel. In the present paper, the threshold crack width is set to 0.1 µm. It should be noted that with this procedure, the effects of cracking on the overall porosity could be overestimated to a certain extent, because all cracks wider than the threshold crack width will be converted back into the voxel configuration as pores, with the size equal to the voxel size (in this case, 1 µm). This could be overcome by either refining the voxel size/lattice mesh (i.e., reducing the voxel size), which would be computationally very expensive, or by creating “partially cracked” voxels. This latter option will be studied in future research.

### 3.3. Ion Transport and Chemical Degradation

#### 3.3.1. Diffusion

In the absence of advective transport, the ion transport in pure liquid, e.g. explicitly represented by capillary pores ($\Omega_{CP}$) in our domain $\vec{x}$, can be described by the following diffusion equation (7):(7)$$\partial_{t}C^{j}\left(\vec{x}\right)=-\vec{\nabla }\cdot \left(-D_{0}\vec{\nabla }C^{j}\left(\vec{x}\right)\right),\;\forall \;\vec{x}\in \Omega_{CP}$$
where $C^{j}$ is the aqueous concentration of j-th chemical species [mol/m^3^] and $D_{0}$ is the diffusivity of chemical species in the capillary pore water [m^2^/s], which is assumed to be the same for all species. In our system, two species Ca and Si are transported. For C-S-H, the reactive-diffusive equation is written as:(8)$$\partial_{t}\phi_{CSH}\left(\vec{x}\right)C^{j}\left(\vec{x}\right)=-\vec{\nabla }\cdot \left(-\phi_{CSH}\left(\vec{x}\right)D_{p,CSH}\left(\vec{x}\right)\vec{\nabla }\;C^{j}\left(\vec{x}\right)\right)+R_{CSH}^{j}\left(\vec{x}\right),\;\forall \;\vec{x}\in \Omega_{CSH}$$
where $D_{p,CSH}$ is the pore diffusivity of chemical species in the C-S-H phase [m^2^/s] and $\phi_{CSH}$ is the porosity of the C-S-H phase. The evolution of $D_{p,CSH}$ is assumed to follow Archie’s relationship in Equation (9):(9)$$D_{p,CSH}=D_{0}{\left(\phi_{CSH}\right)}^{n}$$
where n is a calibration parameter determined as 7.23 by Patel et al. [51] on the assumption that the evolution of diffusivity of C-S-H occurs in a similar way as it evolves during hydration. The initial porosity of the C-S-H phase can be computed from the total porosity based on the model proposed by Hansen [83] and reducing it by the capillary porosity $\phi_{\mathrm{cp}}$ at the initial state as:(10)$${\phi_{CSH}|}_{t=0}=\frac{1}{{\omega_{CSH}|}_{t=0}}\frac{w/c-0.17\alpha }{w/c+0.32}-{\phi_{cp}\;|}_{t=0}\;$$

$\omega_{CSH}$ is the fraction of the C-S-H phase in cement, commencing from the initial state of a cement paste microstructure, and α is the degree of hydration. The solution of ion transport in the cement microstructure is modeled by use of the lattice Boltzmann (LB) method. In this study, we employed a two-relaxation-time (TRT) method, which is a variant of the LB method [84]. This formulation has good stability and accuracy in media with contrasting properties and also has a good computational performance, which is comparable to the single-relaxation-time method. TRT comprises the following symmetric ($f_{i}^{+})\;$ and anti-symmetric parts ($f_{i}^{-}$) of the particle distribution functions: (11)$$f_{i}^{+}=\frac{f_{i}+f_{-i}}{2}\;\;\&\;\;f_{i}^{-}=\frac{f_{i}-f_{-i}}{2}$$
where $f_{i}$ and $f_{-i}$ refer to the distribution functions corresponding to lattice direction $e_{i}$ and opposite to lattice direction $e_{-i}$, respectively. The propagation of the di stribution function according to the TRT-LB equation for the j-th species in the case of multi-component reactive transport is given as: (12)$$f_{i}^{j}\left(\vec{x}+e_{i}\Delta t,t+\Delta t\;\right)=f_{i}^{j}\left(\vec{x},t\right)+\Delta t\;Q_{i}^{TRT,j}\left(\vec{x},t\right)+\Delta t\;Q_{i}^{rxn,j}\left(\vec{x},t\right)$$
(13)$$Q_{i}^{TRT,j}\left(\vec{x},t\right)=-\frac{1}{\tau_{+}}\left(f_{i}^{+,j}\left(\vec{x},t\right)-f_{i}^{eq+,j}\left(x,t\right)\right)-\frac{1}{\tau_{-}}\left(f_{i}^{-,j}\left(\vec{x},t\right)-f_{i}^{eq-,j}\left(\vec{x},t\right)\right)\;$$
(14)$$Q_{i}^{rxn,j}\left(\vec{x},t\right)=w_{i}\;R^{j}\left(\vec{x},t\right)\;$$
where $f_{i}^{eq}$ is the equilibrium distribution function. $\tau_{+}\;$ and $\tau_{-}$ are relaxation parameters for the symmetric and anti-symmetric part, respectively. $R^{j}$ represents a source–sink term from homogeneous chemical reactions [mol/(m^3^s)], and $w_{i}$ are weights, which depend on the lattice type. For the D3Q7 element, the weights $w_{i}$ are $w_{0}=\frac{1}{4}$ and $w_{1-6}=\frac{1}{8}$. Mass conservation is assured through the zero moment of $f_{i}\left(x,t\right)$ along with the summation over the equilibrium distribution, resulting in a concentration of the j-th specie:(15)$$\underset{i}{\sum }f_{i}^{j}=C^{j}$$

The above formulation is valid for diffusion in a pure medium, such as in explicitly represented pores, i.e., pores that are larger than 1 μm. Non-resolved pores, i.e., pores smaller than the discretization size of 1 μm, are conceptualized as porous media. In the LB model, this approach is employed to describe transport in the C-S-H phases. The geometry is characterized by a porosity ϕ and diffusion by an effective diffusion through C-S-H. In this case, the formulation for porous media is defined [85] with the equilibrium function: (16)$$f_{i}^{eq,j}=\frac{C^{j}}{2}\;\left(c_{\phi }+\phi {\vec{e}}_{i}\cdot {\vec{u}}_{i}+I_{n}\phi {\vec{e}}_{i}{\vec{e}}_{i}:{\vec{u}}_{i}{\vec{u}}_{i}\right)\;\forall \;i=1,\ldots ,6\;f_{0}^{eq,j}=\phi C^{j}-\overset{6}{\underset{i=1}{\sum }}f_{i}^{j}$$
where $c_{\phi }$ is the positive adjustable parameter, which is defined by a diffusion coefficient within a stable region, and with the relaxation parameter $\tau_{-}$ as:(17)$$D_{p}=\frac{c_{\phi }}{\phi }\left(\tau_{-}-\frac{\Delta t}{2}\right)$$
and where $c_{\phi }$ has to be set constant throughout the domain to ensure a correct recovery of Equation (7) [51]. As an inlet boundary, i.e., the boundary where leaching occurs, deionized water conditions with fixed concentration are employed. This is implemented as: (18)$$f_{i}^{j}=c_{\phi }C^{j}-f_{-i}^{j}$$

All other boundaries are set to the zero-flux boundary condition, which is implemented using a bounce back scheme, where the unknown incoming $f_{i}$ at a given node is set equal to the outgoing $f_{i}$ in the opposite lattice direction.
(19)$$f_{i}^{j}=f_{-i}^{j}$$

The evolution of the pore structure due to dissolution/precipitation of non-diffusive mineral phases is employed using static update rules [51]. Specifics on implementation of the static update rules for the dissolution of portlandite, and the porosity update of the C-S-H phase due to dissolution, are presented in the next section. Portlandite and clinkers are considered as non-porous materials with zero diffusion. Relative effective diffusivity and pore connectivity were calculated according to the numerical procedures described in [45,47,48]. 

#### 3.3.2. Numerical Leaching

In the present example, the largest phases, i.e., portlandite and C-S-H, are explicitly taken into consideration, while other minor hydration products are treated as C-S-H, in general, and added to the existing C-S-H. In addition, the influence of ion activity and electro-kinetic charges on diffusion was neglected, leading to the simplification that all species have the same diffusion coefficient. This leads to the reactive-diffusive equation for C-S-H as already defined in Equation (8). In this equation, $R_{CSH}^{j}$ represents the source/sink term due to chemical reaction of the C-S-H phase [mol/(m^3^·s)]. This reaction term is calculated from a difference between the actual concentration after diffusion and equilibrium concentration with time according to:(20)$$R_{CSH}^{j}\left(\vec{x}\right)=\phi_{CSH}\left(\vec{x}\right)\partial_{t}\left(C^{j}\left(\vec{x}\right)-C_{eq}^{j}\left(\vec{x}\right)\right),$$
where $C_{eq}^{j}\left(\vec{x}\right)$ is the equilibrium concentration of the j-th chemical species [mol/m^3^]. For C-S-H dissolution, $C_{eq}^{Ca}$ and $C_{eq}^{Si}$ depend on the $C^{Ca}/C^{SI}$ ratio of the C-S-H phase. These two relations are abstracted from a geochemical reaction model by using thermodynamic parameters from CEMDATA07 and treating C-S-H as an ideal solid solution. The results are stored as a look-up table, which is queried using a second-order Lagrange polynomial interpolation function, as detailed in [53]. Portlandite is considered to be a non-diffusive phase. Its dissolution is assumed to be an equilibrium reaction, which entails that the dissolution kinetics is transport-driven. The reaction term for portlandite is defined as:(21)$$R_{CH}\left(\vec{x}\right)=\partial_{t}\left(C^{j}\left(\vec{x}\right)-C_{eq}^{j}\left(\vec{x}\right)\right),$$

This volumetric source of portlandite is applied through the boundary flux between portlandite and either a capillary pore interface or a portlandite C-S-H interface. In those cases where there is contact with a non-diffusive and non-reactive phase (e.g., clinker), a zero-flux boundary condition is employed.

#### 3.3.3. Acceleration of Computational Time for Numerical Leaching

As mentioned previously, dissolution processes are, by nature, very slow processes. However, the process in a cementitious system can be significantly numerically accelerated because of the high chemical buffering (= high ratio between solid concentration *C_solid_*/equilibrium concentration *C_eq_*) of Ca in CH and C-S-H phases and Si in C-S-H phases. This acceleration is performed by a reduction of the necessary number of time iterations, as explained in [20]. A valid reduction of time steps is evaluated on the basis of buffering numbers (Bu) defined as:(22)$$Bu=\frac{V_{total}\cdot C_{eq}}{V_{solid}\cdot C_{solid}},$$
where *V_solid_* is the volume of the solid phase and *V_total_* is the total volume. The ratio *V_solid_/V_total_* should be more than 10%, as shown in [20], and this condition is respected for all solid phases in the hardened cement paste. In order to keep the error of the resulting volume of leached phases below 5%, the acceleration can be performed up to Bu=1, which is sufficiently accurate for the purpose of this analysis.

Considering the material used in this work (Table 1), the volume fraction of C-S-H is around 70% and CH around 10% by volume. The Bu number for Si in C-S-H is then ($C_{eq,Si}=3.1\times 10^{-5}\frac{mol}{dm^{3}},\;C_{Si,\;CSH}=2.42\frac{mol}{dm^{3}}$)
(23)$$Bu_{Si,\;CSH}=\frac{\left(1-V_{CH}\right)\times C_{eq}}{V_{CSH}\times C_{solid}}=\frac{\left(1-0.1\right)\times 3.1\times 10^{-5}}{0.7\times 2.42\;}=1.6\times 10^{-5}$$

For Ca in initial C-S-H ($C_{eq,Ca}=1.92\times 10^{-2}\frac{mol}{dm^{3}},\;C_{Ca,\;CSH}=3.93\frac{mol}{dm^{3}}$):(24)$$Bu_{Ca,\;CSH}=\frac{\left(1-V_{CH}\right)\times C_{eq}}{V_{CSH}\times C_{solid}}=\frac{\left(1-0.1\right)\times 1.92\times 10^{-2}}{0.7\times 3.93}=6.2\times 10^{-3}$$
and for Ca in CH ($C_{eq,Ca}=1.92\times 10^{-2}\frac{mol}{dm^{3}},\;C_{Ca,\;CH}=3.86\frac{mol}{dm^{3}}$):(25)$$Bu_{Ca,\;CH}=\frac{\left(1-V_{CSH}\right)\times C_{eq}}{V_{CH}\times C_{solid}}=\frac{\left(1-0.7\right)\times 1.92\times 10^{-2}}{0.1\times 3.86}=1.5\times 10^{-2}$$

The time step can be increased in such a way that the least buffering phase (in this case, Ca in CH) reaches unity (to assure 5% accuracy) as:(26)$$t_{scaled}=t_{original}\frac{1}{Bu_{original}}=t_{original}\times \frac{1}{1.5\times 10^{-2}}=t_{original}\times 66.6$$

This means that the allowable acceleration is limited to 66 times. However, in order to keep the error as small as possible, an acceleration of 50 times was employed. The acceleration is done by increasing equilibrium concentrations 50 times (could be done by decreasing solid concentration 50 times). The equilibrium concentrations of portlandite remains constant and, hence, for the accelerated conditions, $C_{eq,Ca}=9.6\cdot 10^{-1}mol/l$. For C-S-H, the equilibrium concentrations change with the stoichiometry of C-S-H, which is shown in Figure 3.

With the employed acceleration, the leaching problem could be solved in a few hours on a regular PC computer, which is significantly faster than without acceleration, requiring a computation time of few months. 

## 4. Results and Discussion

As this work involves coupled numerical experiments, results and discussions are divided into the following steps: (1) Microstructure generation, (2) mechanical degradation by external tension load, and (3) chemical leaching of cement paste phases with deionized water.

### 4.1. Generated Microstructures

The hydration simulation was run until obtaining a degree of hydration that provided a good agreement with the reference experimental results, especially with respect to the amount of capillary porosity, portlandite, and C-S-H, where the final volume of C-S-H implicitly mimicked the additional volume of aluminate hydrates (see Table 2).

As the kernel of the Hymostruc model was tuned to match the capillary porosities at which de-percolation starts and ends, a good representation was achieved between these critical points and the dominant transport pathway, which changed from a connected capillary network (parallel capillary/C-S-H mode) into the serial connections of capillary pores entrapped by C-S-H gel pores during hydration. This also allowed for better predictions of the relative diffusivity, where morphological improvements were implemented based on the hollow shell mode of particle expansions (Figure 4). In particular, for particles smaller than the critical diameter, calibrated to 20 μm here [57] (thus named H20), it provided a good match between the calculated pore de-percolation value (50%) with real material [48,49]. By contrast, the reference (unmodified) virtual microstructure exhibits a highly overestimated capillary pore connectivity of 90.3%. Therefore, in the newly proposed hollow shell algorithm, the volume of the inner C-S-H hydration products inside the particles was relocated to the outer shell (inspired by 3D nanotomography measurements [57]), and the resulting vacant space was considered capillary pore space, representing the hollow shell approach, described in the microstructure generation section. In this way, an improved Hymostruc enabled the simulation of a more accurate spatial distribution of capillary and C-S-H gel pores, creating variable closed (ink-bottle) porosities surrounded by small C-S-H gel pores. The reference microstructure is named H0, while H20 stands for the modified numerical microstructure where the number 20 represents the size of the critical particle below which the hollow shell mode of the expansion algorithm was applied. It should be stressed that further research is required here to calibrate and validate this novel hydration modeling approach, as this paper focuses only on testing the possible effect of the resulting pore connectivity on leaching degradation kinetics (which, in fact, turned out to be almost negligible).

### 4.2. Fracture Generation

For the two virtual microstructures H0 and H20, stress/strain curves are obtained from the uniaxial tension simulations with the Delft lattice model. In our model, the loading rate is considered quasi-static (time invariant). Simulation results are plotted in Figure 5, and characteristic numerical data, achieved from the simulations, are summarized in Table 3. In this table, the results are also compared to those from Zhang et al. [66], which were obtained from µCT experiments. From the stress/strain curves, it can be seen that the stiffness, i.e., elastic modulus, for both simulated microstructures is almost the same. This indicates that the heterogeneity of the microstructure is not strongly affected by the activation of the hollow shell model. It also shows that the elastic properties mainly depend on the relative amounts of solid and pore phases in the hardened cement paste, and less on the pore connectivity and the pore structure. Moreover, the strength [77] and post-peak behavior of the two simulated microstructures clearly differ. 

In general, the strength of cement paste is a highly stochastic parameter that highly depends on the (local) heterogeneity of the microstructure [66]. Furthermore, as pores and voids may act as crack initiation sites in quasi-brittle materials, differences in pore size distributions will most likely result in different strengths, even at a constant total porosity. In addition, it can be seen that the improved microstructure (H20) shows a more brittle behavior after the peak load compared to the initial microstructure (H0). This can probably be attributed to the hollow shell implementations in the microstructural model. After crack initiation, cracks tend to propagate throughout the weakest links in the cementitious system. Herein, this means that microcracks will tend to evolve around the capillary pores. However, as the pores in the improved microstructure are less percolated, less distributed microcracking will occur before one or two major cracks start to localize and grow. This can be observed in Figure 6. Two points for plotting the cracks are selected: 0.19% is in the softening regime, in the descending branch, while 0.31% is also in the softening regime, but later, at the beginning of the tail of the curve [60]. In the microstructure H0, significant microcracking occurs and a relatively wide crack zone forms, while a relatively straight crack is formed in the improved microstructure (H20).

### 4.3. Leaching Simulation

Leaching simulations are performed for four microstructures, i.e., initial H0 uncracked, H0 cracked at a strain of 0.31%, initial H20 uncracked, and H20 cracked at a strain of 0.31%. Comparing the diffusion results of the different microstructures in Figure 7 shows that the H20 pore size distribution, which has a lower connectivity with the same porosity than the H0 microstructure, has a two-times lower relative diffusion than the H0 microstructure. In addition, the relative diffusivity increased for the cracked samples, because there is a preferential path formed in the microstructure, and the percolation degree increases. The largest ratio between the diffusivity of the different microstructures is almost three times, viz., between the cracked H0 and the initial uncracked H20 microstructure. With leaching, the diffusivity increases, because cement paste phases are leached out and there is less material present in the leached area. However, in this case, the increase in diffusivity is only considered at the scale of 100 μm and the results should be considered with some caution. After reaching the final simulation time, the dissolution front is at approximately the middle of the domain. This means that only 50 μm of material is still present. The increase in diffusivity would be less sensitive if more material was initially present, e.g., if the initial domain was larger. In addition, if the leaching front progressed too far, this could have led to a violation of representative elementary volume (REV). 

For all four microstructures, a calcium and silica leaching simulation was conducted according to the procedure described above. As the leaching front is not homogeneous (see Figure 8 for Si profile (a) and Ca profile (b)), a procedure for detecting the progressing leaching front had to be developed. This was done by making use of the fact that a significant amount of Si is released from C-S-H, when the Ca/Si ratio is around 1 (this also indicates a pH around 10). 

The dissolved Si diffuses on one side in the direction of the degraded part, and on the other side, into the matrix toward the opposite side of the microstructure. This movement of Si creates a clear peak at the degradation front as can be observed from the red band in the cross-sectional plane in Figure 8a. This peak in Si has been used to determine the leaching front. Numerically, this peak was located by identifying the maximum second derivative of the averaged Si concentration locations. First, we calculate the averaged concentration in each slice according to:(27)$$\bar{C_{Si}}=\frac{\int_{\Omega }^{}C_{Si}d\Omega }{\int_{\Omega }^{}d\Omega }$$
where Ω is an area of a slice perpendicular to the leaching direction. From these averaged values, the actual position was calculated from
(28)$$x_{max,Si}=max\left(\frac{d^{2}\bar{C_{Si}}\left(x\right)}{dx^{2}}\right)$$
with *x* being the position perpendicular to the degradation front.

The calculated positions of the degradation front are plotted in Figure 9, and it can clearly be seen that the propagation of the degradation front is almost similar for all microstructures. See further explanation under general discussion.

### 4.4. General Disscusion

The present research shows an advanced distributed multiscale modeling approach for simulating calcium leaching of aged and degraded concrete surfaces. A set of coupled numerical experiments, involving microstructure generation, mechanical degradation by external tension, and calcium leaching of cement paste phases with deionized water, were conducted and evaluated. Two different virtual microstructures (w/c = 0.5) were generated having the same volume of phases, but a different pore connectivity. The reference microstructure showed a high capillary pore connectivity of 90.3%, while a modified one was tuned to exhibit only half of the capillary pores connected. This change in connectivity was introduced by employing a hollow shell algorithm to those particles with a critical diameter smaller than 20 µm, which led to a shift in the capillary volume from the open capillary volume toward the closed hollow shell space. It also led to a slight increase in the growing-shell expansions of the smaller particles in the hydrating system, and a reduction in the connected capillary pore volume. The modified microstructure resulted in a better agreement with the measured relative diffusivity, which would have been overestimated by using the classical growing sphere model. In addition, each of the initial microstructures were numerically mechanically degraded. This mechanical degradation resulted in the formation of a simulated fracture pattern through the virtual microstructure. The size of the fracture was such that the porosity did not change considerably, but it caused a preferential path through the microstructure. This was also demonstrated by the calculation of the relative diffusivity, where fractured samples showed a significantly higher diffusivity than the non-fractured samples. However, despite this difference in effective diffusivity, the influence on the progress of the leaching front remained almost negligible for all samples. The reason is that the system is chemically sufficiently buffered by solid phases, i.e., solid concentrations of Ca and Si are much larger than the equilibrium concentrations of dissolved Ca and Si. As a result, the concentration profile in the pore structure and micro fractures remains similar. In other words, there is no concentration gradient in pores of the microstructure and, hence, the effective diffusivity of a pore structure has limited influence.

## 5. Conclusions

In this work, we described the modeling chain, which allows for the evaluation of mechanical and chemical degradation of different cement microstructures. The chain consists of three different models, starting with the microstructure generation on which two consequent degradation mechanisms, mechanical and chemical, are applied. Based on the results from these models, the following conclusions could be drawn:A good agreement between modeled and measured relative diffusivity, which would have been overestimated by using the classical growing sphere model, has been achieved by the modification of the virtual microstructure. The modification was based on the hollow shell model of hydration by calibrating the critical particle diameter to obtain a good match between the calculated and expected capillary pore connectivity according to the literature results. Further, research is needed to validate this novel hydration modeling approach, which was used here only preliminarily to test the effect of pore connectivity on leaching degradation kinetics;The controlling mechanism for leaching dissolution kinetics is the amount of soluble phases. The leaching is defined through porosity, because this defines the amount of reactive phases, i.e., larger porosity leads to less solid phases and, thus, less buffering capacity. However, physical parameters, such as pore connectivity and diffusivity, turned out to be negligible in the kinetics of the leaching process;Microcracks turned out to have no significant influence on the leaching process, as long the cracks remain sufficiently small and this size is significantly larger than that of which the present model allows the evaluation. The reason for this is that leaching of hydration products (Ca, Si) highly buffers the pore solution, which overrides the diffusion and pore connectivity role because there is no concentration gradient within the cement matrix and microcracks;With the “numerical experiment,” it was demonstrated that the coupled numerical model enables the possibility to construct a virtual material with desired properties and to show the correlations between their physical properties, i.e., pore connectivity and morphology;The coupled numerical model demonstrated its acceleration potential for simulating degradation processes in aged concrete. This is in comparison to real leaching experiments, where accelerated tests can be achieved by using, for example, ammonium nitrate instead of deionized water, which does not guarantee the same leaching kinetics ratio of all minerals compared to that of ionized water.The uniaxial tensile simulations conducted for the two microstructures showed similar results as reported in previous studies. It was confirmed that the simulated Young’s moduli are governed by the total capillary porosity and the amount of solids in the microstructure, and that it is relatively independent of the pore structure and the spatial distribution of individual solid phases. The simulated tensile strength and the post-peak behavior turned out to be influenced by the pore size distribution and the distribution of hydration phases.

On the basis of these findings obtained on pure hardened cement paste, future research will focus on the influence of aggregates and porous interfacial transition zones (ITZ) in mortars and concrete. 

## Figures and Tables

**Figure 1 materials-13-02697-f001:**
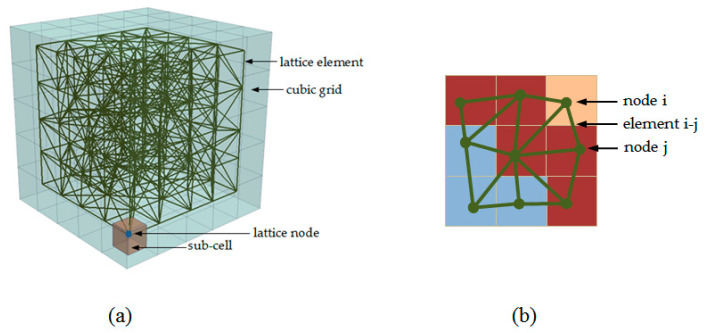
(**a**) Node and mesh generation procedure; (**b**) an example of the overlay procedure for cement paste, shown in 2D for simplicity (pink—outer product, red—inner product, light blue—unhydrated cement), adapted from [77].

**Figure 2 materials-13-02697-f002:**
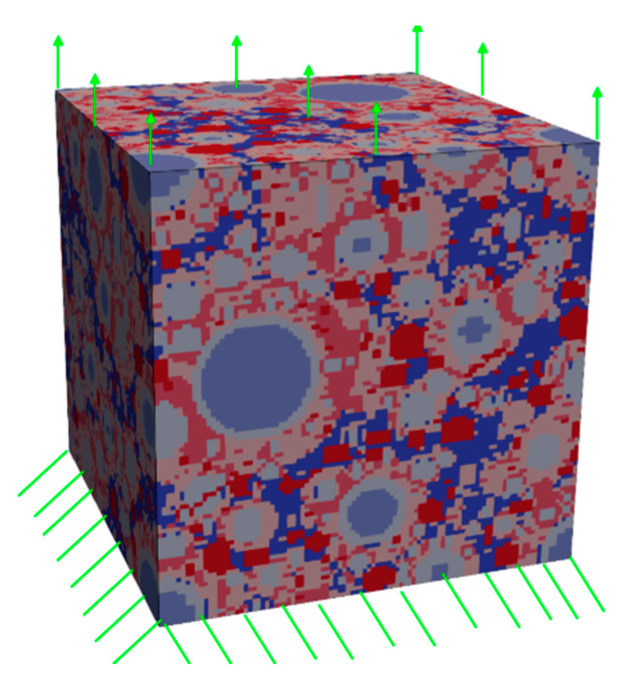
Example of the microstructure and the boundary conditions in the simulated uniaxial tensile test.

**Figure 3 materials-13-02697-f003:**
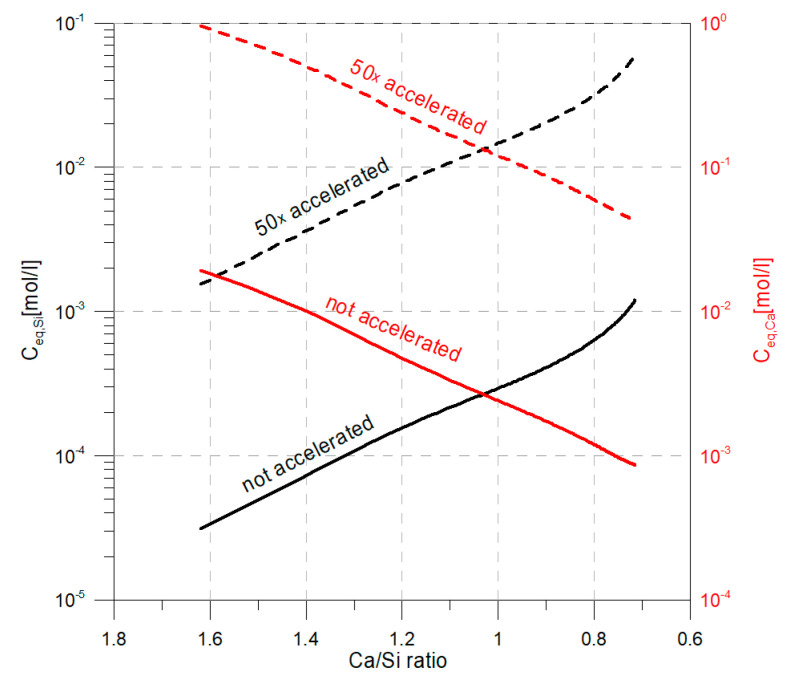
Equilibrium concentrations for accelerated and non-accelerated conditions.

**Figure 4 materials-13-02697-f004:**
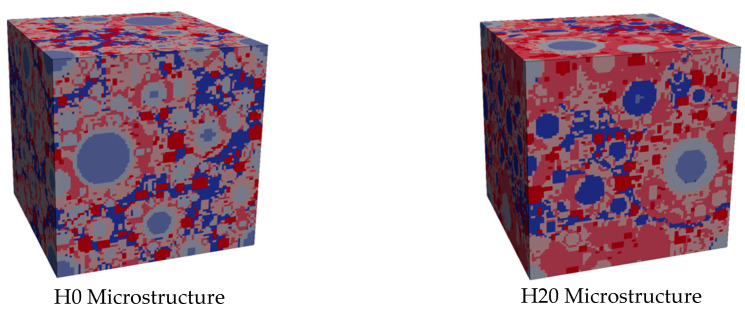
Impression of the two virtual microstructures employed in this research.

**Figure 5 materials-13-02697-f005:**
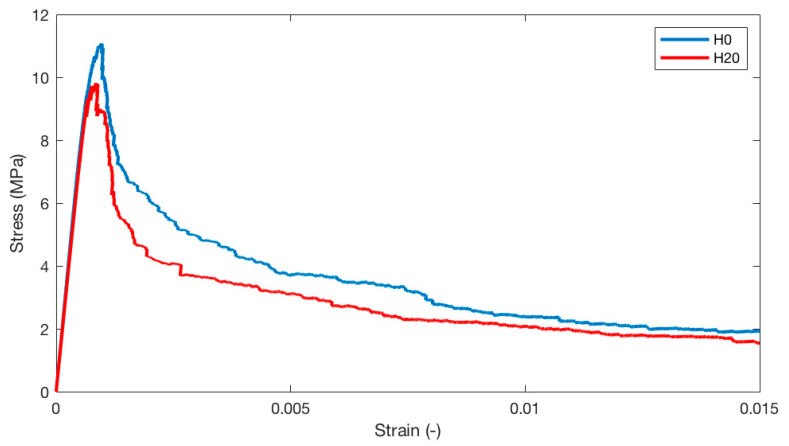
Simulated stress/strain curves for cement paste microstructures subjected to simulated uniaxial tension.

**Figure 6 materials-13-02697-f006:**
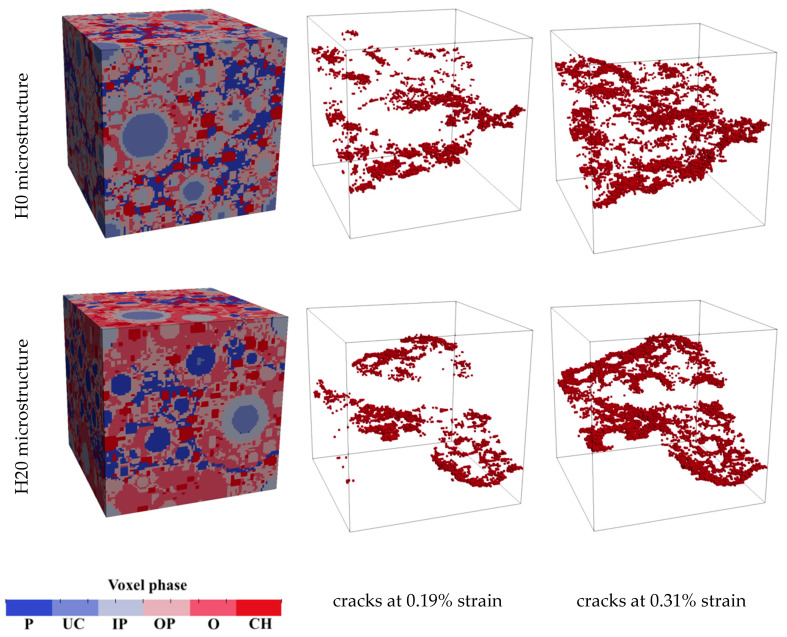
Simulated microstructures and crack development for the two cases. In the legend: P-pore; UC-unhydrated cement; IP-inner product; OP-outer product; O-overlap; CH-calcium hydroxide.

**Figure 7 materials-13-02697-f007:**
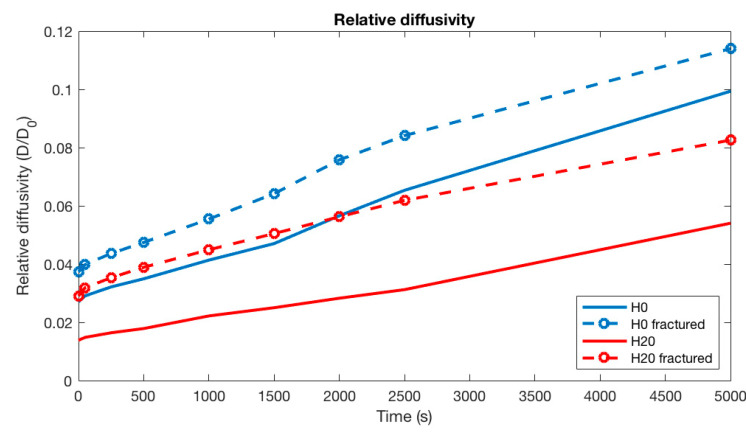
Evolution of the diffusion coefficient during leaching at the scale of 100 µm domain.

**Figure 8 materials-13-02697-f008:**
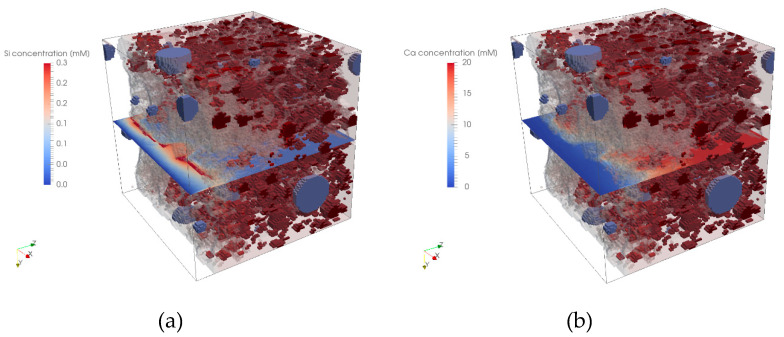
Dissolved Si concentration (cross-section) at the degradation front (**a**) and dissolved Ca concentration cross-section (**b**).

**Figure 9 materials-13-02697-f009:**
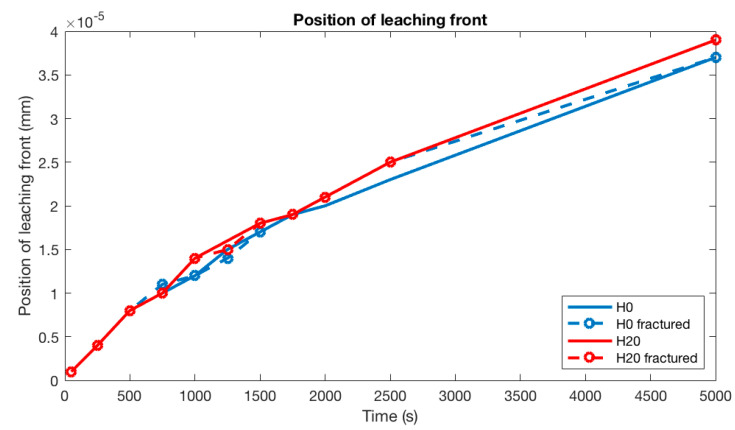
Position of degradation front for the simulated microstructures.

**Table 1 materials-13-02697-t001:** Volume fractions of cement paste.

Composition	Vol %				
Cement phases	clinker	C-S-H	Portlandite (CH)	C-A-H	Capillary porosity
Ngala & Page [56]	3.4	50.0	10.6	20.0	15.7
Hymostruc (H0)	6.0	67.6	10.8	0	15.6

**Table 2 materials-13-02697-t002:** Mechanical properties of different types of lattice elements used in the analysis. Elastic moduli (E) are taken from nanoindentation measurements [79], while tensile strengths ($f_{t}$) are taken from inverse analyses performed in [81]. Note that, in [81], calcium hydroxide and overlap phases were not considered. Herein, calcium hydroxide (CH) is taken to have the same mechanical properties as the inner product [82], while the overlap is taken to be the same as the outer hydration product.

Element Type	Phase i	Phase j	E (GPa)	$f_{t}\;(\mathrm{MPa})$
1	unhydrated cement	unhydrated cement	99	683
2	inner product	inner product	31	92
3	outer product	outer product	25	58
4	overlap	overlap	25	58
5	calcium hydroxide	calcium hydroxide	31	92
6	unhydrated cement	inner product	47	92
7	unhydrated cement	outer product	40	58
8	unhydrated cement	overlap	40	58
9	unhydrated cement	calcium hydroxide	47	92
10	inner product	outer product	28	58
11	inner product	overlap	28	58
12	inner product	calcium hydroxide	31	92
13	outer product	overlap	25	58
14	outer product	calcium hydroxide	28	58
15	overlap	calcium hydroxide	28	58

**Table 3 materials-13-02697-t003:** Uniaxial strength and elastic modulus of simulated microstructures determined by micromechanical simulations. For comparison, results from our previous work are given (note that, there, the microstructure was determined using X-ray CT).

Microstructure	Strength (MPa)	Elastic Modulus (GPa)	Capillary Porosity (%)
H0	11.08	15.60	15.6
H20	9.82	14.84	15.8
Zhang et al. [66]	15.97 ± 5.02	19.36 ± 5.03	15.7 ± 5.76 *

* only pores larger than the resolution of the X-ray CT (2 µm).

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
