# Peer review of "Influence of Micro-Pore Connectivity and Micro-Fractures on Calcium Leaching of Cement Pastes—A Coupled Simulation Approach"

_materials, 2020, doi:10.3390/ma13122697_

Round 1
Reviewer 1 Report
Dear authors,
The paper with title "Influence of micro-pore connectivity and micro-fractures on calcium leaching of cement pastes – a coupled simulation approach"
is interesting and offers new knowledge in the field of the influence of the microstructure of cementitious materials on calcium leaching.
However, it should be rewritten in a clearer way to be published and some corrections should be provided, such as:
1. page 1, line 35: "leaching of concrete is an extremely slow process". Leaching of Calcium ions wille be maybe suitable.
page 2, line 62: Can you transform the sentence "Nevertheless, porosity still increases for long time (more than 2-week leaching)." to preferable form
for better understanding ?
page 2, line 71 - the literature reference is missing for this sentence
In the paragraph (line 72-87) do you mean in all sentences leaching by using deionized water or by another medium ? please specify
page 3, line 109. "curve of calcium in ammonium nitrate solution", you should more specify...do you mean "calcium leaching in ammonium nitrate solution" I suppose
line 119 - "24%" instead of "24 %"
line 131 - "is provided" instead of "will be provided"
line 136 - can you explain what represents "reference one and a modified one" ?
Physical framework:
line 145 - What exactly means "mass ratio of 0.5"
Subsection 2.1. - Why did you use this cement composition for your research ?
line 159 - you should firstly define abbreviation FIB-SEM at the first occasion
line 179 - please transform the sentence---what dou you mean by "based microstructure"
2.3. Chemical degradation - I would like to ask if you think that Ca leaching from etringite and CaCO₃ (calcite, etc.) is also possible ?
line 196 - Can you specify Rosin-Rammler distribution ?
line 204 - you should use experessions C-S-H and C-A-H, because aluminate hydrates do not belong to C-S-H.
line 211 - By the abbrev. "CH" you mean portlandite ? If yes, you should define it firstly.
Table 1 - Is it alright the value of tensile strenghts of unhydrated cement 683 MPa? It is very confusing, try to check
line 270 - You are writing about H0 and H20, but you did not mentioned it before.
Why you did not consider also C-A-H in the calculations ? (3.3.3. Acceleration of computational time for numerical leaching)
Table 2 - marking "Pcap" should be firstly explained, then used.
Figure 4 should be mentioned in the text.
Simulated stress/strain curves shoould be connceted with the testing time or loading rate should be mentioned.
Table 3 - Why is the deviation for Zhang et al. [45] so large ? Any deviations should be add also to H0 and H20 or there was not ?
Figure 6 - Why did you choose strain 0.31% and 0.19% ? It should be explained.
Figure 7 - scale is missing
Figure 8 - Dissolved Si concentration (cross-section) is stated, but do you have similar image for Dissolved Ca concentration ?
Best regards
Author Response
The authors thank the reviewer for thorough review which enabled us to improve our manuscript. We made our best to address the reviewers’ comments in a sincere and correct manner. However, we could not resolve one request related to the Figure 7 (one before last, marked with red colour). We don’t find any missing scale in the figure. We kindly ask the reviewer for more detailed description of the missing scale if the comment is still relevant. Thank you.
Our replies are given below.
Point 1: 1. page 1, line 35: "leaching of concrete is an extremely slow process". Leaching of Calcium ions wille be maybe suitable.
Response 1: Indeed, we mean the Ca-leaching. Text is corrected.
Point 2: page 2, line 62: Can you transform the sentence "Nevertheless, porosity still increases for long time (more than 2-week leaching)." to preferable form for better understanding ?
Response 2: Thank you for your suggestion. We have revised the sentence.
Point 3: page 2, line 71 - the literature reference is missing for this sentence
In the paragraph (line 72-87) do you mean in all sentences leaching by using deionized water or by another medium ? please specify
Response 3: Reference added. Yes, this paragraph deals with the leaching in deionized water. We slightly modified the sentences to make them clear.
Point 4: page 3, line 109. "curve of calcium in ammonium nitrate solution", you should more specify...do you mean "calcium leaching in ammonium nitrate solution" I suppose
Response 4: The original text is correct. This is to refer to the solid-liquid equilibrium curve of calcium in ammonium nitrate solution, which is different from Ca equilibrium curve in deionized water. And indeed this curve is sued in case of calcium leaching in ammonium nitrate solution as the reviewer mentioned.
Point 5: line 119 - "24%" instead of "24 %", line 131 - "is provided" instead of "will be provided"
Response 5: Text corrected
Point 6: line 136 - can you explain what represents "reference one and a modified one" ?
Response 6: The previous sentence has been modified to: Firstly, the pore connectivity of the original Hymostruc model (reference microstructure) was modified by implementing an innovative hollow shell growing sphere approach (modified microstructure), which reduced the connectivity to a more realistic one
Point 7: line 145 - What exactly means "mass ratio of 0.5"
Response 7: The w/c expression was introduced to make this clear:
‘…a cement and water mixture with a mass ratio of w/c = 0.5, …’
Point 8: Subsection 2.1. - Why did you use this cement composition for your research ?
Response 8: Following text was added in this section: ‘This particular cement paste composition was used to represent the experimental results from [38], considered to be a valuable reference for measured effective diffusion coefficient.
The chemical degradation process described in this research is based on calcium leaching from an Ordinary Portland Cement (OPC, i.e. CEM I) based microstructure. The modelling approach proposed can be used to simulate the effects of other microstructural parameters, e.g. w/c, chemical composition, mineral additions (i.e. different cement types), chemical additives etc., however we believe this would not change the conclusions of this paper.’
Also, Table 2 is now shifted forward as Table 1 in subsection 3.1., where the simulation of the virtual microstructure is introduced: ‘The hydrated virtual cement paste microstructures were simulated by the Hymostruc model representing the experimental results from [38], and summarized in Table 1.’
Point 9: line 159 - you should firstly define abbreviation FIB-SEM at the first occasion
Response 9: The focused ion beam scanning electron microscope (FIB-SEM) was defined, in this line where it was introduced for the first time.
Point 10: line 179 - please transform the sentence---what dou you mean by "based microstructure"
Response 10: The sentence is rephrased to: The chemical degradation process described in this research is based on a microstructure representing an Ordinary Portland Cement (OPC).
Point 11: 2.3. Chemical degradation - I would like to ask if you think that Ca leaching from etringite and CaCO₃ (calcite, etc.) is also possible ?
Response 11: The model allows for leaching of any chemical system. However, the leaching of ettringite of calcite requires additional chemical components and consequent calculation by geochemical solver. In our model the chemical system is simple and we could make a tabulated dependencies from Ca/Si ratio. The extension to more complex systems is under developments (also for precipitation).
Point 12: line 196 - Can you specify Rosin-Rammler distribution ?
Response 12: Following part was added to specify the Rosin-Rammler distribution, and more importantly the calculation of the number of particles per fraction:
‘The Rosin-Rammler distribution is defined as:
where G(d) is the cumulative weight (in g) of the particles with diameter (in μm), and parameters n and b are constants defined by the fineness of the cement. The number of particles in a certain fraction are calculated from the weight per fraction, obtained by differentiating eq 1 with respect to the particle diameter d, as well as by dividing this by the specific density of the cement (ρ) and by the volume of a single particle (Vp = π d3 / 6):
‘
Furthermore, Hymostruc model description was briefly introduced, as shown in respond to Review No 2.
Point 13: line 204 - you should use experessions C-S-H and C-A-H, because aluminate hydrates do not belong to C-S-H.
Response 13: We have added C-A-H expression for aluminate hydrates. This is also implemented in Table 1.
Point 14: line 211 - By the abbrev. "CH" you mean portlandite ? If yes, you should define it firstly.
Response 14: The definition of CH was introduced timely. This is also implemented in Table 1.
Point 15: Table 1 - Is it alright the value of tensile strenghts of unhydrated cement 683 MPa? It is very confusing, try to check
Response 15: The value is correct. Hydration phases do show exceptionally high strength on the micro-scale, in part due to the size effect. It has been determined by inverse modelling in our previous work (Zhang, Hongzhi, et al. "Microscale testing and modelling of cement paste as basis for multi-scale modelling." Materials 9.11 (2016): 907.). This is in line with experiments directly performed on the micro-meter sized cantilever beams by Němeček, J., et al. "Tensile strength of hydrated cement paste phases assessed by micro-bending tests and nanoindentation." Cement and Concrete Composites 73 (2016): 164-173.
Point 16: line 270 - You are writing about H0 and H20, but you did not mentioned it before.
Why you did not consider also C-A-H in the calculations ? (3.3.3. Acceleration of computational time for numerical leaching)
Response 16: (We assume C-A-H means CH) CH was not directly used, because to date we do not have reliable calibrated values for its tensile strength.
Point 17: Table 2 - marking "Pcap" should be firstly explained, then used.
Response 17: Pcap was replaced with Capillary porosity
Point 18: Figure 4 should be mentioned in the text. Simulated stress/strain curves shoould be connceted with the testing time or loading rate should be mentioned.
Response 18: Figure 4 is now referenced in the text. Sentence added for explanation: In our model the loading rate is considered quasi-static (time invariant).
Point 19: Table 3 - Why is the deviation for Zhang et al. [45] so large ? Any deviations should be add also to H0 and H20 or there was not ?
Response 19: In the present work, two simulations are performed: one for H0 and one for H20. Therefore, there are no deviations. In the work of Zhang et al [45], 30 simulations with different microstructures are performed and their results compared. Given the heterogeneity of cement paste at this scale, the deviation is reasonable.
Point 20: Figure 6 - Why did you choose strain 0.31% and 0.19% ? It should be explained.
Response 20: The corresponding explanation is given: Two points for plotting the cracks are selected: 0.19% is in the softening regime, in the descending branch, while 0.31% is also in the softening regime, but later, at the beginning of the tail of the curve
Point 21: Figure 7 - scale is missing
Response 21: Authors kindly ask to point out precisely what is missing. In this graph, there is a legend and both axes have scale.
Point 22: Figure 8 - Dissolved Si concentration (cross-section) is stated, but do you have similar image for Dissolved Ca concentration ?
Response 22: Additional figure and its reference in the text with Ca profile is added.

Reviewer 2 Report
materials-830710 - Review Report:
The paper is focused on a very interesting and current topic influence of micro-pore connectivity and micro-fractures on calcium leaching of cement pastes by a coupled simulation approach.
In this work we described the modelling chain, which allows for the evaluation of mechanical and chemical degradation of different cement microstructures. The chain consists three different models, starting with the microstructure generation on which two consequent degradation mechanisms, mechanical and chemical, are applied.
The paper has sufficient scientific interest and has originality in its technical content to merit publication. The paper is also structured well, and I believe the topic of the paper would be of interest to the journal's audience. The article is written with great care for rhetoric and text editing.
The paper deserves to be published with only one minor modifications.
1) Lines 55-57- “Nevertheless, the level of modifications strongly depends on the compositions of the native materials. In general, cementitious materials with less portlandite content exhibit a better leaching resistance, and the same trend is observed for materials having pozzolanic activity.” - it would be suggested to include a reference for these statements.
2) Line 144 - In the study, w/c = 0.5 was used ("... a cement and water mixture with a mass ratio of 0.5" - should this be understood as the w / c ratio?) Would the model be used for another w/c ratio value?
3) What type of cement was modeled? There is a lot of information in the introduction that the type of cement and mineral additives (micro and nano) matter.
4) In practice, the concrete in its composition contains chemical additives that affect the porosity. How does this relate to your research? That is why it is so important that the model has a similar composition to universal concrete used in practice.
5) Only one from “References” is from 2019, it is worth providing more recent reports (reference) regarding research.
6) Line 159 - FIB-SEM - it is worth explaining the acronym.
7) Line 132 - reference is made to [37]. This is a study for the Hymostruc model. Model Hymostruc is the acronym for HYdration, Morphology and STRUCtural development - I find, but in the article I did not find this explanation. In [37] the background and structure of the simulation model are discussed.
The authors do not describe the Hymostruc model, citing only literatureHowever, a brief presentation of the model should be attached, the more so that the model is modified by implementing an innovative hollow shell growing sphere approach.
8) It is difficult to find an explanation of the H0 and H20 cases under consideration.
Author Response
The authors thank the reviewer for relevant comments and suggestions. We made our best to address the reviewers’ comments in a sincere and correct manner. Our replies are given below.
Point 1: Lines 55-57- “Nevertheless, the level of modifications strongly depends on the compositions of the native materials. In general, cementitious materials with less portlandite content exhibit a better leaching resistance, and the same trend is observed for materials having pozzolanic activity.” - it would be suggested to include a reference for these statements.
Response 1: References are added as suggested by the reviewer.
Point 2: Line 144 - In the study, w/c = 0.5 was used ("... a cement and water mixture with a mass ratio of 0.5" - should this be understood as the w / c ratio?) Would the model be used for another w/c ratio value?
What type of cement was modeled? There is a lot of information in the introduction that the type of cement and mineral additives (micro and nano) matter.
In practice, the concrete in its composition contains chemical additives that affect the porosity. How does this relate to your research? That is why it is so important that the model has a similar composition to universal concrete used in practice.
Response 2: The w/c expression was introduced to make this clear:
‘…mimicking a cement and water mixture with a mass ratio of w/c = 0.5, …’
Following sentence was added:
‘This particular cement paste composition was used to represent the experimental results from [ref], considered to be a valuable reference for measured effective diffusion coefficient.
The chemical degradation process described in this research is based on calcium leaching from an Ordinary Portland Cement (OPC, i.e. CEM I) based microstructure. The modelling approach proposed can be used to simulate the effects of other microstructural parameters, e.g. w/c, chemical composition, mineral additions (i.e. different cement types), chemical additives etc., however we believe this would not change the conclusions of this paper.’
Point 3: Only one from “References” is from 2019, it is worth providing more recent reports (reference) regarding research.
Response 3: Thanks for your suggestions, we added few more recent references (2019, 2020) including our view on those studies in the Introduction section. However, it’s worth noting that there are not many studies on Ca-leaching in recent years. In the years from 2010-2019, the peak of Ca-leaching studies has been reached and now it’s slowing down. Also some new references related to the modelling are added.
Point 4: Line 159 - FIB-SEM - it is worth explaining the acronym.
Response 4: The focused ion beam scanning electron microscope (FIB-SEM) is now defined in this line.
Point 5: Line 132 - reference is made to [37]. This is a study for the Hymostruc model. Model Hymostruc is the acronym for HYdration, Morphology and STRUCtural development - I find, but in the article I did not find this explanation. In [37] the background and structure of the simulation model are discussed.
The authors do not describe the Hymostruc model, citing only literature. However, a brief presentation of the model should be attached, the more so that the model is modified by implementing an innovative hollow shell growing sphere approach.
Response 5: Following brief presentation of the Hymostruc model was introduced in Section 3.1 and 3.2:
The Rosin-Rammler distribution is defined as:
|
|
where G(d) is the cumulative weight (in g) of the particles with diameter (in μm), and parameters n and b are constants defined by the fineness of the cement. The number of particles in a certain fraction are calculated from the weight per fraction, obtained by differentiating eq 1 with respect to the particle diameter d, as well as by dividing this by the specific density of the cement (ρ) and by the volume of a single particle (Vp = π d3 / 6):
|
|
Section 3.2:
For cement reactions the following reaction equations are considered:
where z = 2 or 3 and represents the silicate reaction for C3S (Alite or tricalcium silicate) and C2S (Belite or dicalcium silicate), respectively. …
In the hydration model Hymostruc, the formation of a 3D microstructure is simulated with the assumption that hydrating cement particles behave like expanding spheres, that follow an integrated kinetics approach according to the so-called ‘Basic Rate Equation’ [ref], which describes the reaction front into an individual reacting particle i with diameter d, at time t:
(4)
In this kinetic equation (4) K0 is the initial rate of hydration reaction of the individual cement particles, FArr is the Arrhenius function for temperature dependence, Wi factors account for the state of water in the microstructure during hydration [ref]. Parameters δtr and δd,j are the transition thickness (a constant threshold value) and the thickness of the total layer of produced hydration products, respectively. The last part in the equation (4) uses a Boolean parameter λ which accounts for the change of the particle reaction mechanism from phase boundary (λ = 0) to a diffusion controlled (λ = 1), and β is an empirical constant [ref]. The incremental evolution of the degree of hydration of cement is then calculated for each individual particle (and for each of the Bogue phases), and is based on the actual depth of the reaction front of a particle i (δd) while Δδd is the incremental increase of this front. The Δδd is calculated for each reaction scheme, and the actual reaction front of a particle is obtained by mass averaging the individual reaction contributions. For a more detailed description of the algorithm reference is made to [ref].
Point 6: It is difficult to find an explanation of the H0 and H20 cases under consideration.
Response 6: The reference to H0 and H20 models are introduced earlier in the text. The following was added: “This adjustment enabled reduction of the actual connected capillary pore volume in the system and the results of this calibration are detailed in the section describing microstructure generation. Two virtual microstructures are generated, which are different in terms of pore connectivity, but the same in terms of porosity and the amount of solid phases. The reference microstructure is named H0, indicating that the hollow shell algorithm is not invoked by setting the value of the threshold particle diameter to 0 mm; as hollow shells are being created for cement particle sizes below this critical value. The second microstructure is modified for particles smaller than the critical diameter, here calibrated to 20 mm [ref] and thus named H20 to provide a good match between the calculated pore de-percolation value with real material. In the newly proposed hollow shell algorithm the volume of the inner C-S-H hydration products inside the particles were relocated to the outer shell (inspired by a 3D nanotomography measurements [ref]) and the resulting vacant space was considered capillary pore space, representing the hollow shell approach. This was implemented in the existing Hymostruc kernel following two major steps. Firstly, the phase tag associated with the inner C-S-H hydration products was changed to a capillary pore (i.e. colour changed from red to blue). Secondly, the implemented algorithm that accounts for the calculation of the expanded C-S-H growth around the particles, was extended with the additional volume coming from the inner C-S-H shell volume, which is equal to the volume of the reacted cement. It was not necessary to change the overlap algorithm of the Hymostruc model, as it implicitly calculates the expansion of particle shells, also when considering hollow shell transformations.”

Reviewer 3 Report
Comments to the Authors:
The authors of this paper present an advanced reactive- transport model where results are used for leaching induced aging. However, some details may be considered by the authors:
COMMENT: Page 1, lines 27-28: I suggest the authors to add more recent references (Several studies have been published in recent years on the effects of pollution and climate change on concrete made civil engineering and cultural heritage structures. Therefore, I suggest the authors to choose and add more characteristic studies, as references).
COMMENT: Page 1, line 37: More recent references could be added (Several studies have been published in recent years about the pH, porosity and transport properties. Therefore, I suggest the authors to choose and add more characteristic studies, as references).
COMMENT: Page 14, line 494: This result may be further commented (the result that the increase in diffusivity with the progression of leaching is similar for all cases may be further commented.For example, why this happens, since the microstructures are different?)."
Author Response
The authors thank the reviewer for relevant comments and suggestions. We made our best to address the reviewers’ comments in a sincere and correct manner. Our replies are given below.
Point 1: Page 1, lines 27-28: I suggest the authors to add more recent references (Several studies have been published in recent years on the effects of pollution and climate change on concrete made civil engineering and cultural heritage structures. Therefore, I suggest the authors to choose and add more characteristic studies, as references).
Response 1: Thanks for your suggestions, we added few more recent references (2019, 2020) including our view on those studies in the Introduction section (lines 71-80).
The effects of pollution and climate change on leaching of concrete is limited as leaching is more relevant in saturated conditions. Such issues is more relevant for carbonation, corrosion which are not the focus of this study. Therefore, we did not review such works. Please let us know if we overlooked some relevant literature in the field.
Point 2: Page 1, line 37: More recent references could be added (Several studies have been published in recent years about the pH, porosity and transport properties. Therefore, I suggest the authors to choose and add more characteristic studies, as references).
Response 2: Thanks for your suggestion, we have added few more recent papers related to the effects of leaching on alteration of microstructure, mineralogy and transport properties. New references are added for the modelling part (lines 140-144)
Point 3: Page 14, line 494: This result may be further commented (the result that the increase in diffusivity with the progression of leaching is similar for all cases may be further commented.For example, why this happens, since the microstructures are different?)."
Response 3: The sentence is deleted at that place, but additional text is given in the “General discussion”: The reason is that the system is chemically sufficiently buffered by solid phases, i.e. solid concentration of Ca and Si are much larger than the equilibrium concentrations of dissolved Ca and Si. As a result, the concentration profile in the pore structure and micro fractures remains similar. In other words, there is no concentration gradient in pores of microstructure and hence the effective diffusivity of a pore structure has a limited influence.

Round 2
Reviewer 1 Report
Dear authors, thank you for the thorough incorporation of all my comments, especially, I appreciate the explanation considering the leaching of Ca from ettringite of calcite. I agree with you and wish you good luck in solving this very interesting issue.
To the Point 21: Figure 7 - scale is missing
I meant, that caption: "Evolution of the diffusion coefficient at the scale of 100 µm - domain during leaching." should be more suitable.
And Fig. 7 should be mentioned also in the text.
After checking and editing the English grammar, I recommend the paper for publication in the journal Materials.
Author Response
Authors thank for the clarification of the point raised by the reviewer
Point 1:I meant, that caption: "Evolution of the diffusion coefficient at the scale of 100 µm - domain during leaching." should be more suitable.
Response 1: The caption have been adapted to: "Evolution of the diffusion coefficient during leaching at the scale of 100 µm - domain"
Also the reference to Fig. 7 has been made. Thank you once again for this careful review.